# The Barthlott effect

Laurent Vonna

Institut de Science des Matériaux de Mulhouse, Université de Haute-Alsace, Université de Strasbourg, Mulhouse, France

## Classics

epicuticular wax; lotus effect; superhydrophobicity.

**Corresponding author:**
Laurent Vonna;
Email: laurent.vonna@uha.fr

## Abstract

In 1997, Barthlott and Neinhuis published a groundbreaking article entitled "Purity of the sacred lotus, or escape from contamination in biological surfaces" that caused a true paradigm shift in surface science. In this article, they explained the water-repellent and self-cleaning properties of plants, attributing the superhydrophobicity to nano- and micrometric wax textures on the surface of the leaves. This became known as the "Lotus Effect". In the late 1980s, Barthlott already demonstrated the microtexture of plant surfaces and its effect on wetting. However, this knowledge remained confined to botany until the 1997 article popularized it. The dissemination of this knowledge to the materials science community led to the development of countless synthetic superhydrophobic surfaces and a better understanding of wetting mechanisms. The story of this discovery and its consequences demonstrates the relevance of atypical approaches and emphasizes the urgency of respecting biodiversity.

In 1997, the journal *Planta* published an article by Barthlott and Neinhuis (1997) that opened up a whole new and still very dynamic field of materials science by focussing attention on the importance of surface texture for wetting. In this groundbreaking article, "Purity of the sacred lotus, or escape from contamination in biological surfaces" (5,162 citations, July 2023, Web of Science—Clarivate), the two botanists propose an explanation not only for the water-repellent character of various plants but also for the self-cleaning effect that this character is linked to. Based on scanning electron microscopy (SEM) images of plant leaf surfaces, it is proposed that water repellency (or superhydrophobicity) arises from texturing of the epicuticular wax at the nano- and micrometric scales (Figure 1a). The extremely reduced interface between the leaf surface and the water caused by this texturing leads to very low adhesion with water and to the rolling of water droplets, that is raindrops, which carry dust or other surface contaminants with them, creating a self-cleaning effect. The phenomenon described and explained in this way was later to be popularized as the 'lotus effect'.

Since that publication, the number of papers referencing the article has multiplied. To give an idea of the extent of the impact of this article, Figure 1b shows the occurrence of the terms 'lotus effect' and 'superhydrophobic', which were popularized after this publication to describe the water-repellent character (Barthlott et al., 2016). The spectacular nature of the phenomenon thus revealed has led many teams, since the early 2000s, to attempt its reproduction with synthetic materials. At the same time, others have followed the approach of Barthlott and Neinhuis using SEM to reveal the nano- and microtexture of the surfaces of other plants, as well as other biological objects, such as insects or animals, that exhibit unique interactions with water. It would be difficult to list all the different combinations of surface textures and materials with superhydrophobic properties that have been proposed since then without extensive bibliographic work. This abundance of work on the phenomenon represents a true paradigm shift in surface science, although it has not necessarily led to remarkable technological applications.

The research in materials science that followed the publication of this article led to the proposal of synthetic surface textures with much higher resistance to wetting by water than those found in nature. Research then quickly turned to the development of surface textures resistant to wetting by liquids with surface tensions lower than that of water, such as oils (superoleophobic surfaces), for which the surface texture of a lotus leaf, for example, does not prevent wetting as observed for water. All these efforts have been accompanied by considerable progress over the last 25 years in understanding the wetting mechanisms of textured surfaces. In some cases, this

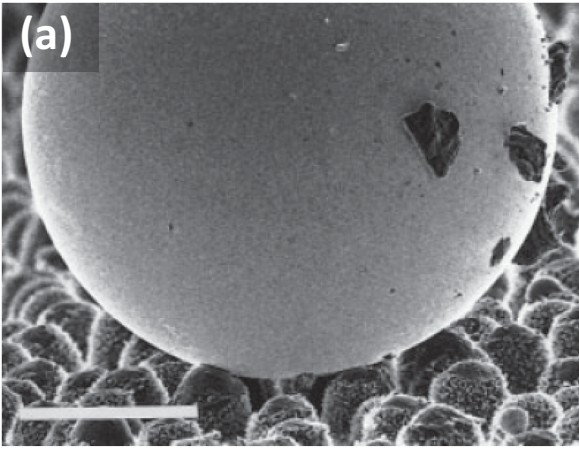

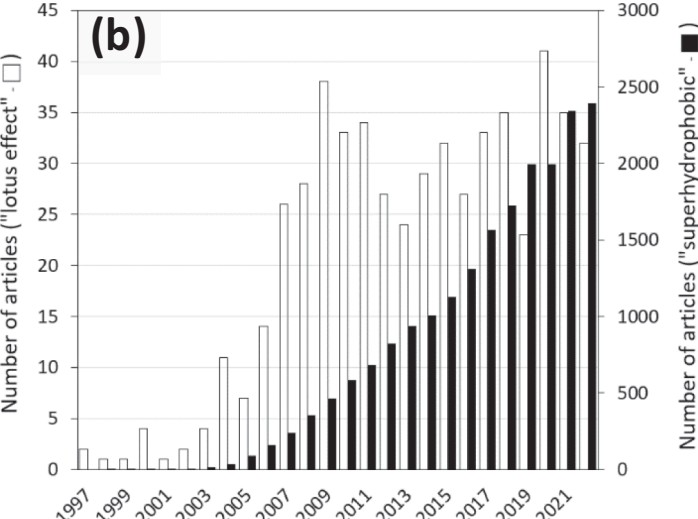

**Figure 1.** (a) SEM image from the article "Purity of the sacred Lotus. . ." showing a mercury droplet on the adaxial leaf surface of *Colocasia esculenta*. The spherical shape of the droplet (also triggered by the high surface tension of mercury) and the contaminating particles adhering to the droplet demonstrate the 'lotus effect' (bar =50 $\mu$m).
(b) Occurrence of the terms 'lotus effect' and 'superhydrophobic' since 1997 according to Web of Science-Clarivate (for the search fields 'Title', 'Abstract' and 'Keywords'). Using this bibliographic tool, the term 'superhydrophobic' already appeared 6 times before 1997 in the field of chemical engineering and cell science.

has necessitated the development of wettability characterization techniques complementary to the commonly used sessile drop technique.

While the effect is indeed spectacular and has a relatively simple explanation, it is remarkable that this article aroused so much interest after its publication. After all, the effect of surface texture on wetting has been described and known for several decades, especially since the famous papers by Wenzel (1936) and Cassie and Baxter (1944). Superhydrophobicity was even observed and described, without naming it as such, by Dettre and Johnson (1965) on rough wax surfaces loaded with glass beads. Later, in 1991, Busscher *et al.* produced a paper whose title includes the term superhydrophobic, in which they describe this behaviour on Teflon surfaces whose surface nanotexture generated by ion ablation produces a water-repellent effect (Busscher et al., 1991). The spread of SEM techniques in the 1970s contributed greatly to the study of surface nano- and microtextures, which had not been possible before. This was precisely the case for Barthlott, who began his career with the application of the technique to the characterization of the surfaces of plant leaves and flowers at these scales at the Institute of Botany at the University of Heidelberg (Neinhuis, 2017). Since the 2000s, the development and popularization of microfabrication techniques for producing ever smaller and more precise surface textures, possibly combined with the popularization of SEM or atomic force microscopy (AFM), may also have contributed to this dynamic.

Interestingly, Barthlott revealed the microtextures of plant surfaces as early as 1977 by publishing SEM images of the epidermal surfaces (shoots, leaves and perianths, seed coats) of nearly 2,100 angiosperms and 45 gymnosperms in a German language botanical journal (Barthlott & Ehler, 1977). He even discusses the 'biological-ecological significance', as he puts it, of these surface textures, and in particular, for our purposes here, how they affect wetting. Barthlott presented the self-cleaning effect more clearly in 1981 (Barthlott & Wollenweber, 1981), citing in passing Cassie and Baxter's (1944) paper, which was purely in the field of materials science and later became famous for discussing the principle of superhydrophobicity. The 'lotus effect', or self-cleaning effect, was even clearly stated in the title of an article by Barthlott published in 1992, "Die Selbstreinigungsfähigkeit pflanzlicher Oberflächen durch Epicuticular wachse" (Barthlott, 1992), which can be translated into English as "The self-cleaning ability of plant surfaces through epicuticular wax", an article still in a botanical journal and still in German language.

These observations, published in confidential German-speaking journals, therefore already contained all the ingredients for the explosion of studies on the 'lotus effect' and the properties of superhydrophobicity in the late 1980s. But, as Neinhuis reports in an article honouring Barthlott (Neinhuis, 2017), the information remained confined to the relatively confidential field of systematic botany that, by his own admission, was considered old-fashioned in the 1970s and 1980s, and characterized by the time-consuming process of patient observation and classification. In the same article, he notes the boundaries between scientific fields, at least at that time, and the difficulty of information dissemination in both the academic and industrial worlds. It was finally only after the publication of their article in the journal Planta in 1997, and a thorough popularization effort, that the concept spread throughout the materials science community.

The perseverance in this work of observation and classification, far from the spectacular race for potential technological innovations triggered by the 'lotus effect', led Barthlott and collaborators, in 2009 and 2010, to reveal in the water fern *Salvinia* a remarkable ability to stabilize an air plastron once submerged (Barthlott et al., 2010; Koch et al., 2009). It is interesting to note that the authors of these articles, perhaps now aware of the boundaries between research fields, formulated and wrote their statements in such a way as to be published in journals edited in the field of materials. In a similar way, but on a smaller scale than the 'lotus effect', this property, which is referred to as the 'Salvinia effect', again aroused the interest of the materials community in reproducing it. The sophisticated surface texture of the leaves of the water fern *Salvinia* consists of complex hydrophobic hairs with hydrophilic end regions. These hydrophilic areas stabilize the air

layer by pinning the air–water interface, resulting in the long-term retention of an air plastron between the hydrophobic hairs. This surface texture associated with an underwater air plastron shows remarkable drag reduction and many other interesting properties relevant to various technologies (Barthlott et al., 2017; Bing et al., 2021). The observation and classification work of Barthlott, Neinhuis, and their colleagues have finally led to the creation of a database on the surface microtexture of several thousand biological objects (freely available at www.lotus-salvinia.de).

It seems to us that the long observation time that characterizes botany also allows fundamental questions to be asked at the edge of the speed and immediacy of applied research. For example, Barthlott and Neinhuis, at the very beginning of the competition for the technological optimization of superhydrophobic surfaces, asked about the molecular mechanisms, in particular crystallization, behind the different morphologies that characterize wax microtextures on the surface of superhydrophobic plants (Meusel et al., 1999; 2000; Neinhuis et al., 2001). They have also published several papers on the self-assembly of epicuticular waxes using AFM (Koch et al., 2003; 2004) and have discussed the influence of air humidity during plant growth on the composition and morphology of waxes and, ultimately, on the wettability of leaf surfaces (Koch et al., 2006). Barthlott and Neinhuis have also investigated the evolutionary pathways that led to superhydrophobicity in plants, in particular as the first organisms transitioned from aquatic to terrestrial life (Barthlott et al., 2016; 2017).

Far from the performance indicators of research, sheltered from the dictates and financial pressures of applied research, patient observation of nature thus made it possible to discover and explain the superhydrophobic property of plant surfaces, which in the end was easy for anyone to observe since, as Barthlott and coworkers estimate, it affects a total surface area of our planet of about 250 million km$^2$ (Barthlott et al., 2017). This landmark discovery in materials science demonstrates not only the relevance of observing and understanding living things but also through the history of its discovery and emergence, the relevance in research of atypical approaches, incongruous associations, on the fringes of fashion, as observed by Neinhuis (2017). The temporality of the development and application of the superhydrophobic property on synthetic surfaces contrasts with the millions of years of evolution of nature towards this property. But Barthlott, in a call to respect the diversity of life, reminds us that this slow evolution also contrasts with the rapid depletion of biodiversity that characterizes our time, diminishing biological models for potential innovations (Barthlott et al., 2017).

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
