## [Reviewer Report]

I have read the article by Lautent Vonna with great pleasure and interest. In his essay entitled “The Bartholott Effect”, the author described an outstanding scientist in the scientific community whose research went far beyond the framework of the scientific discipline he presented.

The paper’s title refers to the Lotus Effect - the phenomenon of self-cleaning of water-repellent leaf surfaces characterized and explained by Batholott and coworkers.

Many discoveries in various scientific fields originate from cognitive curiosity about the world and careful observation of the surroundings. Some are consequences of a systematic and patient nature-spotting, while others result from accidental discovery. In the author’s view, Barhlott constitutes the scientist who does not take a shortcut. He pointed out that Barholott’s thorough research into the nature of the Lotus phenomenon began with systematic botany (images of epidermal surfaces over 2000 angiosperm and gymnosperm), followed by a molecular mechanism of wax crystalization and its differential morphology, which is the essential feature for lotus effect. The fate of scientific discovery is often influenced by time and place (thematic scope of the magazine in which it is published) of its publication, as the author suggested. The Barholott case is an apparent example of a milestone discovery in natural science in basic research, which triggered applied research in an entirely distinct scientific field. Furthermore, his study resulted in appearance of many followers from the materials science community, which indicates a significant softening of the boundaries between distant scientific disciplines. It’s worth mentioning that the widespread use of the Internet in the 21st century as a source of scientific information and its exchange makes for the dissemination of knowledge in both the academic and industrial worlds. Simultaneously, multidisciplinary team formation facilitates engaging between basic and applied research and favours the implementation of natural solutions, which have evolved for millions of years into potential innovations.

---

## [Reviewer Report]

A well-balances article on the history and implications of the discovery of the Lotus Effect - and the non-conventional approach that led to this discovery.

---

## [Reviewer Report]

I would like to apologize for the very long time that have passed from the paper submission. The reason is that finding the Reviewers turned out to be really diffucult. Nevertheless, now I am pleased to say that both the Reviewers, and also myself, are recommending acceptance of the manuscript.